# Chronic Myeloid Leukemia in Children: Immune Function and Vaccinations

**DOI:** 10.3390/jcm10184056

**Published:** 2021-09-08

**Authors:** Meinolf Suttorp, Andrea Webster Carrion, Nobuko Hijiya

**Affiliations:** 1Hematology and Oncology, Medical Faculty, Technical University, D-01307 Dresden, Germany; 2Division of Pediatric Hematology, Oncology and Stem Cell Transplantation, Columbia University Medical Center, New York, NY 10032, USA; aw3240@cumc.columbia.edu (A.W.C.); nh2636@cumc.columbia.edu (N.H.)

**Keywords:** chronic myeloid leukemia, CML, tyrosine kinase inhibitor, immunizations, COVID-19

## Abstract

Children with CML need TKI treatment for many years, and the lack of knowledge about immune dysfunction with TKI has hindered routine immunizations. This review attempts to provide an overview of the effects of TKIs licensed for children (e.g., imatinib, dasatinib, and nilotinib) on immune function, as well as its implications on immunizations. We discuss surveillance strategies (e.g., immunoglobulin blood serum levels and hepatitis B reactivation) and immunizations. All inactivated vaccines (e.g., influenza, pneumococcal, and streptococcal) can be given during the treatment of CML in the chronic phase, although their efficacy may be lower. As shown in single cases of children and adults with CML, live vaccines (e.g., varicella, measles, mumps, rubella, and yellow fever) may be administered under defined circumstances with great precautions. We also highlight important aspects of COVID-19 in this patient population (e.g., the outcome of COVID-19 infection in adults with CML and in children with varying hemato-oncological diseases) and discuss the highly dynamic field of presently available different vaccination options. In conclusion, TKI treatment for CML causes humoral and cellular immune dysfunction, which is mild in most patients, and thus infectious complications are rare. Routine immunizations are important for health maintenance of children, but vaccinations for children with CML on TKI therapy should be carefully considered.

## 1. Introduction

Chronic myeloid leukemia (CML) is a clonal myeloproliferative malignancy characterized by the presence of a BCR-ABL1 fusion gene as a consequence of the t(9;22)(q34.1;q11.2) reciprocal chromosomal translocation. CML is a rare disease in children and adolescents, with an estimated annual incidence of 2.5 cases per million in children and young adults and accounting for 2–3% of all childhood leukemia cases and ~9% of leukemia cases in adolescents between 15 and 19 years of age [1,2,3]. The current standard of care for patients with CML is indefinite tyrosine kinase inhibitor (TKI) treatment, while discontinuing TKI treatment is possible in a subset of patients [4]. TKIs function by blocking the activity of BCR-ABL1. The introduction of TKIs has dramatically increased the survival of patients with CML. However, TKIs inhibit not only BCR-ABL1 but also many other targets, and they cause various side effects via off-target effects, including impaired immune function [5]. Children with CML need TKI treatment for many years, and the lack of knowledge about immune dysfunction with TKIs has hindered routine immunizations.

In this review, we provide an overview of the effects of TKIs on immune function that have emerged to date, as well as its implications for immunizations. We discuss surveillance strategies and immunizations. We also highlight important aspects of COVID-19 in this patient population and discuss the different vaccination options.

## 2. Effect of TKI on Immune Function

BCR-ABL1-specific TKIs that are used for the treatment of CML are not entirely BCR-ABL1-specific and inhibit other targets (e.g., c-KIT, TEC, SRC, FLT3, Lck, and mitogen-activated kinases (MAPK)). This “off-target” effect causes various adverse events and alters immune responses. The immunosuppressive effects of TKIs have been demonstrated in vitro and in animal models by modulating the differentiation of dendritic cells (DCs) as well as by impeding proper T-cell responses and macrophage functions [6,7]. Patients with CML have impaired innate and adaptive immunity at diagnosis, and patients on TKI treatment are considered to be clinically vulnerable. While data on the immune function in children receiving TKIs for CML are lacking, opportunistic infections or serious infectious complications are not reported in large pediatric CML trials [8,9,10]. Rohon et al. investigated the immunoprofile at diagnosis of CML and during therapy with imatinib and dasatinib in adult patients (*N* = 54) [11]. A lower proportion of B cells and dendritic cells and an increased number of NKT-like cells were observed in the BM at diagnosis. With imatinib therapy, all these changes returned to normal, and the immunoprofile was similar to the healthy controls. Among patients receiving dasatinib, however, two groups were identified. One group resembled healthy controls, while the other group showed activation of immune functions characterized by significant elevations of CD8+ and NK- and NKT-like cells. Data on immune functions in patients on other TKIs are sparse [12].

### 2.1. Altered Humoral and Cellular Immune Function

The plasma immunoglobulin levels were measured in adult patients with CML at diagnosis and after 12 months of treatment with imatinib (*N* = 20), dasatinib (*N* = 16), nilotinib (*N* = 8), and bosutinib (*N* = 12) [13]. The proportion of patients with IgA, IgG, and IgM levels below the lower limit of normal was 0%, 11%, and 6%, respectively, at diagnosis; however, at 12 months, they increased to 6% (*p* = 0.13), 31% (*p* = 0.042), and 28% (*p* = 0.0078), respectively. Low IgG levels in imatinib-treated patients were associated with higher percentages of immature bone marrow B cells, and IgG levels in the low-to-normal range at diagnosis in patients predisposed them to hypogammaglobinemia during the treatment with TKIs. Patients who had low immunoglobulin levels during the TKI therapy experienced significantly more frequent minor infections during the follow-up compared with the patients with normal levels (33% vs. 3%, *p* = 0.0016). Another study showed that TKIs, particularly second-generation TKIs with greater off-target kinase inhibition, inhibited B-lymphocyte functioning and the antibody response [14].

Similar findings were reported in a pediatric cohort with CML (*N* = 30, mean age: 16.4 years, range: 9–23 years) where they were treated with imatinib at a median dose of 287.5 mg/m^2^ (IQR: 267.3, 345.0) for a median duration of 6 years [15]. The IgG, IgA, and IgM plasma levels decreased in 9 (30%), 8 (27%), and 10 (33%) patients, respectively. In 5 (17%) patients, pan-hypogammaglobulinemia was detected. Thus, if patients with CML on imatinib experience recurrent or unusual infections, measuring the immunoglobulin levels should be considered.

In general, infectious complications are rare in patients with CML of all ages who are on TKI treatment. Routine prophylaxis for opportunistic infections (e.g., pneumocystis jirovecii) is not recommended [16]. However, during the first months after diagnosis, leukopenia as a side effect of TKI treatment is observed commonly in children. In this situation, short-term pneumocystis prophylaxis may be considered in addition to temporary interruption of TKI treatment. TKIs may cause reactivation of CMV infections, with a particularly higher risk for dasatinib treatment [17,18]. Depending on the geographic region, patients may also benefit from screening for tuberculosis and treatment of latent infections [19,20].

A risk of hepatitis B reactivation in patients with CML on TKI therapy has been reported, while hepatitis C infections have not been reported so far [21]. However, there are no clear guidelines or recommendations regarding the screening and monitoring of hepatitis B virus (HBV). Atteya et al. reviewed the literature to estimate the risk of chronic HBV reactivation associated with TKI treatment and addressed the following unanswered questions: (1) is there a need to screen all patients who will receive TKIs; (2) how long should patients with CML be on HBV prophylaxis, and (3) which are the best antiviral agents to use for prophylaxis against HBV for patients on TKIs [22]? The authors concluded that (1) it is advisable to screen all patients scheduled for TKI treatment by testing with ALT, a complete hepatitis B serology panel (including HBsAg, anti-HBs antibodies, anti-HBc antibodies, HBeAg, anti-HBe antibodies), and HBV DNA, (2) in HBsAg positive patients, prophylaxis against HBV reactivation should be started, as reactivation may happen any time after the commencement of TKIs with a median time of 9–10 months (range: 1–69 months), and (3) there was no clear answer from the literature as to which antiviral drug should be used.

### 2.2. Immunizations

Very little data are available about the safety and efficacy of vaccinations for immunosuppressed patients [23]. As a consequence, immunocompromised children are under-vaccinated and vulnerable to vaccine-preventable infections. It is very difficult to study the efficacy of vaccinations for rare diseases such as pediatric CML due to the very small sample size. Biological parameters demonstrating a protective effect in healthy individuals may not be extrapolated to immunocompromised individuals. Lower responses to vaccination in immunosuppressed individuals compared with healthy people are expected, and little data exist on the durability of the response. Concerning live vaccines, with a few exceptions, these are generally considered to be contraindicated in immunosuppressed individuals because of safety concerns.

The prevalence of children with CML is constantly increasing as the disease becomes more curable. Stem cell transplantation is associated with a high risk for morbidity but has become a third line option, and it is performed for very few children. TKI treatment may be needed for many years, and that makes the timing of vaccination crucial. Two different goals must be achieved by vaccination in immunosuppressed children: (1) protect the patient against specific infections whose risks are evidently increased by the treatment in comparison with healthy individuals and (2) offer an individual patient the same protection as the healthy community (e.g., against measles or influenza) [24].

#### 2.2.1. Inactivated (Killed) Vaccines

Inactivated vaccines are generally safe in immunosuppressed patients, but it must be considered to what extent they are efficacious. Mild adverse events, mostly local reactions at the injection site, may be observed, but they are similar to immunocompetent hosts. Data in the literature regarding vaccination in immunosuppressed patients are scarce. However, the guidelines from the 2017 European Conference on Infections in Leukaemia (ECIL 7) for patients with hematological malignancies including CML have recently been published [24]. In the guidelines, TKIs inhibiting BCR-ABL1 are categorized as only mildly immunosuppressive, and all inactivated vaccines can be given during treatment, although their efficacy may be lower.

Vaccination with an inactivated influenza vaccine is recommended annually. It was demonstrated in adults that the proportion of patients exhibiting a protective immune response (antibody titer 1:32 or more) was slightly inferior in patients with CML (85%) compared with the healthy controls (100%). However, T-cell responses to the H1N1 vaccine were not significantly different between the patients and controls [25].

Patients with CML should also be vaccinated against *Streptococcus pneumoniae*. It has been demonstrated that TKIs impair B-cell responses through off-target inhibition of the kinases involved in cell signaling. Adult patients with CML on TKIs achieved lower pneumococcal IgG and IgM titer responses (75% versus 100% in the healthy controls) after vaccination with PPSV23 [14]. The immunogenicity of pneumococcal conjugate vaccines (PCVs) is higher than PPSV23 because of the T-cell dependent response induced by the conjugation with the diphtheria protein. While there are no data on PCV in patients with CML, one may reasonably deduce from data in other immunocompromised populations to recommend one dose of PCV, followed at least 2 months later by one dose of PPSV23 [24].

Other inactivated vaccines should be administered according to the guidelines in each country. Current issues concerning vaccination against COVID-19 (all developed vaccines thus far are killed vaccines) are discussed below in a separate section. It must be kept in mind that immunoglobulin titers are low in a considerable proportion of children on TKI treatment. In the pre-TKI era, it was demonstrated that 18% of patients were not immune to tetanus [26]. Thus, it may be prudent to check the titers against inactivated vaccines regularly and to perform a booster vaccination in children with non-protective levels of titers. It should be also considered that the expected response rates to vaccines may be lower when on dasatinib or bosutinib (not yet licensed in children) treatment than with imatinib or nilotinib [14].

#### 2.2.2. Live Attenuated Vaccines

According to current recommendations, vaccinations with live vaccines are completed by the age of 4–6 years in most countries. As CML is rarely seen in children below that age, only very few children face the issue of live vaccine administration during TKI treatment [27,28,29]. A problem exists for older children who missed their booster vaccinations, as well as an increasing number of children whose parents generally have refused vaccination (e.g., the measles vaccine) [27,28,30].

Live vaccines are typically contraindicated in immunosuppressed individuals in general. While the degree of immunosuppression may vary among patients receiving antineoplastic treatment, emerging data support the safety and effectiveness of live vaccines in certain immunocompromised individuals. In Swiss travel clinics, 197 patients on immunosuppressive treatments including corticosteroids, mesalazine, methotrexate, and TNF-alpha inhibitors received live vaccines against yellow fever, measles/mumps/rubella (MMR), varicella, or oral typhoid vaccines. In this cohort, no serious side effects or infections by the attenuated vaccine strain occurred [31].

For children with CML on TKI therapy, live attenuated vaccines should be administered with great caution. For patients who are in deep molecular response, a window for vaccination can be created by interrupting the TKI therapy [27]. Experience in children with and without interruption of TKI treatment is limited so far to vaccinations against MMR and VZV in four patients [30]. When a patient is on treatment with imatinib (no experience exists with the other TKIs), live vaccines may be considered on an individual basis if the following prerequisites are fulfilled: (1) the patient lives in or is traveling to an area where a vaccine-preventable infection is endemic, (2) the patient is in the chronic phase of CML and in an overall stable situation (a complete cytogenetic response has been achieved with a BCR-ABL1/ABL1 ratio of 1% or lower (expressed on the international scale), peripheral blood lymphocyte counts stable at >1500/μL, and changes in the full blood count caused by imatinib treatment or switching to another TKI not expected), (3) a prior vaccination with an inactivated vaccine has resulted in an adequate immune response, and (4) the benefits and risks of the planned live attenuated vaccination are discussed in depth with the patient and his or her legal guardians [30].

##### Attenuated Varicella Virus Live Vaccine

The natural course of a varicella zoster virus infection has been described in a cohort of adult patients on imatinib, in which 16 out of 771 patients (2%) developed VZV infections (15 episodes of herpes zoster and 1 of varicella). All patients received and responded well to therapy with antiviral agents. The authors concluded that imatinib therapy in CML is associated with a low incidence of VZV infection, does not disseminate, responds well to therapy, and does not mandate a recommendation for herpes zoster prophylaxis in patients with CML [32]. In pediatric trials of imatinib, no unusual cases of VZV infection were reported [8,9,33]. This is in contrast to the experience with other pediatric cancers or other diseases requiring immunosuppressive therapies, which have the risk for a potentially fatal course of VZV infection. A varicella live attenuated vaccine may be given to patients with CML [30]. Our own very limited experience with attenuated live VZV vaccination stems from two girls aged 14 and 15 years with CML-CP and under imatinib treatment for 2 years and 3 years, respectively, having achieved major molecular response when vaccinated. Both patients tolerated the vaccination well without any side effects, but only one girl developed protective serological VZV titers [30]. As a word of caution, the zoster live attenuated vaccine, which contains 15–20-fold higher titers of attenuated viruses than the varicella live attenuated vaccine, should not be used, as no experience with the zoster live vaccine has been published thus far.

##### Attenuated Measles Mumps Rubella Live Vaccine

The measles virus is considered to be one of the most highly contagious known human pathogens [34]. The World Health Organization’s (WHO’s) goal of global elimination of measles by vaccination successfully prevented an estimated 21 million deaths worldwide since the year 2000. However, despite this achievement, there is concern of a new increase in the number of measles cases reported globally [35]. The growing number of travel-related infections and local outbreaks in industrialized countries due to vaccine refusal is alarming. Endemic measles has now been reestablished in several European countries where transmission was previously dormant. Additionally, with more than 1000 reported measles cases from 1 January 2019 to 20 June 2019, the United States experienced the largest number of measles cases per annum since the disease was eliminated in the year 2000 [36]. Aside from North America and Europe, the seriousness of the problem is highlighted by the WHO’s report of 207,500 measles deaths worldwide in 2019, the highest annual number of deaths since 1996, occurring mostly in countries with weak health systems [37,38]. For the last few years, the priority has shifted to handling the current pandemic of COVID-19, and millions of children are at risk of not receiving measles vaccines [39]. Thus, patients with CML undoubtedly will need protective serum titers not only when traveling to countries where measles are endemic but also when a local outbreak occurs in their area.

As outlined before, most children diagnosed with pediatric CML are at school age, and protective serum titers against measles should be the same as in the age-matched general population. In Germany, 88.8% of the children were MMR-vaccinated at least once, and a study with more than 13,000 children and adolescents aged 0–17 years found that 76.8% of them showed evidence of antibodies to MMR [40].

To what extent pediatric patients with malignancies lose their humoral immunity against vaccine-preventable diseases was investigated in a German single-center study comprising 195 children (122 male) with ALL (*N* = 80), AML (*N* = 15), non-Hodgkin’s lymphoma (*N* = 18), Hodgkin’s disease (*N* = 22), and various solid tumors (*N* = 60). Overall, 27%, 47%, 19%, and 17% of the patients lost their humoral immunity against measles, mumps, rubella, and VZV, respectively [41]. To the best of our knowledge, no such analysis has been performed for CML under TKI treatment so far.

Bettoni da Cunha-Riehm et al. published their very limited experience with live attenuated vaccines for pediatric CML, including four patients aged 12–15 years who were on imatinib treatment for 1–3 years and had missing protective measles titers during local outbreaks of measles in Germany [30]. After careful consideration of the risks and benefits, three patients were vaccinated while receiving TKI therapy, while imatinib treatment was interrupted in the fourth patient for 1 week prior and 2 weeks after vaccination. No acute or late adverse events from vaccination were observed in any of the four patients. While patients 1 and 3 developed stable long-term seroconversions, a serum titer conversion against measles and varicella could not be demonstrated in patient 2. However, MMR revaccination, given 3 years later, did not result in the development of a protective measles titer or the titer was lost again. Patient 4 also had lost protective titers against measles when assessed 10 months after the first vaccination, but revaccination resulted in stable seroprotective titers that were stable for over 12 months during a follow-up. Of note, no clear conclusions should be drawn from these four cases until more experience from a larger number of patients demonstrates that live attenuated vaccines are safe and that stable protective titers are achieved. Whether or not TKI impedes the seroconversion by blocking attenuated virus proliferation via a blockade of the virus’ release from an infected cell (see below in “Putative Antiviral Action of TKIs”) should also be investigated in more detail [38,42,43].

##### Yellow Fever Live Attenuated Vaccine

Yellow fever (YF) is endemic in the tropical regions of Africa and South America. The course of an infection is serious, with case fatality rates of 20–50%. A very effective live attenuated vaccine was developed in the 1930s, and a single vaccination providing lifelong immunity has proven to be critical in the control of the disease [44]. An international certificate of vaccination may even be required when entering a country from another region where yellow fever is endemic [45]. Travel medicine authorities may advise immunosuppressed patients to avoid yellow fever endemic regions altogether or provide letters of exception when there is an entry requirement but little or no actual risk of exposure [46].

Vaccination against YF is recommended in children from the age of 9–12 months onward. However, a decline in specific immune response has been observed as a consequence of a lower seroconversion rate observed in infants compared with adults. Booster regimens should be performed to guarantee the long-term persistence of immune protection for children living in areas with a high risk of YF transmission [47].

Concerning patients on immunosuppressive treatment, the question arises as to what extent the protection achieved against YF by vaccination is maintained. One smaller study examined 35 healthy individuals as controls and 40 immunosuppressed adult patients (autoimmune diseases and organ transplantation), all having received YF vaccination prior to the onset of their immunosuppression [48]. With a median follow-up interval of 21.1 years (interquartile range: 14.4–31.3) after YF vaccination and while taking immunosuppressive drugs, no statistical difference was found, exhibiting a total of 35 seropositive immunosuppressed patients (88%) compared with 31 patients (89%) in the control group.

Another report focused on the side effects of vaccination with a YF live attenuated vaccine when it was administered inadvertently to 19 immunosuppressed patients (prednisone, azathioprine, cyclosporine, mycophenolate, sirolimus, or tacrolimus) following solid organ transplantation (kidneys *N* = 14, heart *N* = 3, and liver *N* = 2). Transplantation had been performed at a median interval of 65 months (range: 3–340) prior to vaccination [49]. None of the 19 patients experienced side effects except for slight reactions at the injection site in one case.

The largest data set on immunosuppressed patients receiving YF vaccines stems from a report describing when, in 2016, the largest outbreak in several decades of YF occurred in a previously transmission-free area in southeast Brazil and expanded to previously YF-free areas in highly populated areas near Espirito Santo, Rio de Janeiro, and Sao Paulo. YF vaccination was expanded to the entire population living in areas without prior vaccine recommendations in which the outbreaks took place. Given the high risk of YF transmission, experts standardized the criteria for YF vaccination to include immunocompromised patients in Sao Paulo. Low-grade immunosuppression was defined by single-drug therapy with hydroxychloroquine or sulfasalazine (any dosage), corticosteroids in a prednisone equivalence dosage of ≤2 mg/kg or ≤20 mg daily, methotrexate ≤0.4 mg/kg or ≤20 mg weekly, or leflunomide ≤20 mg/day. The YF vaccine was defined as contraindicated for persons taking methylprednisolone pulse therapy, mycophenolate mofetil, cyclosporine, cyclophosphamide, azathioprine, JAK inhibitors, or biological immunomodulators [50], and 381 immunosuppressed individuals (median age: 50.8 years, range: 1.4–89.3 years) without prior YF vaccination were vaccinated with a full dose of the YF 17DD vaccine. Although more details are not explicitly listed in the report, among these were 12 patients with hematological malignancies, including patients with CML, of whom 5 were on imatinib treatment and 2 were on dasatinib. From the total cohort of immunocompromised vaccinees, at least one adverse event was reported by 32.6% of patients, with no statistically significant difference in the spectrum of complaints according to the vaccine producer report. Four severe events, including 3 deaths, were observed but did not occur in patients with CML and were classified to be not related to the vaccination. The authors concluded that the YF attenuated live vaccine may be administered to mildly immunocompromised, clinically stable persons living in high-risk areas, always preceded by a careful individual assessment weighing the benefits and risks of vaccination.

To the best of our knowledge, no further data on YF vaccination in CML are available at this time. Therefore, we consider it desirable to put efforts on this issue in countries with emerging financial resources, where YF is endemic and where patients with CML have access to TKI treatment. In addition, data from YF vaccination in patients on TKI might support approaches with other live attenuated vaccinations such as MMR and varicella.

### 2.3. COVID-19

#### 2.3.1. Putative Antiviral Action of TKIs

During the COVID-19 pandemic, data reported from Italy and China pointed to a lower prevalence in patients under TKI treatment. Some authors argued that this finding might demonstrate a protective effect of TKI therapy [51,52,53]. Earlier in vitro experiments had demonstrated that the tyrosine kinase ABL1 is involved in controlling the protein arrangement of the cytoskeleton. If inhibited by imatinib in infected cells, the syncytia formation induced by the corona virus spike protein is blocked [54,55]. Using quantitative assessment of the torque teno virus (TTV) viremia as a model for virus replication in immunosuppressed patients and patients with CML on imatinib, it was shown that in contrast to other immunosuppressive drugs, the TTV load did not increase while on TKI treatment [56,57,58]. From a clinical viewpoint, two trials, one in the Netherlands and one in France, will evaluate the benefits of early imatinib therapy to prevent severe COVID-19 disease in adult patients (COUNTER-COVID, ClinicalTrials.gov (accessed on 31 August 2021) Identifier NCT04357613) [59,60].

#### 2.3.2. COVID-19 Infection in Patients with CML

In China, 530 patients with CML at a median age of 44 years (range: 6–89 years) were studied by questionnaire during the COVID-19 epidemic in Hubei Province [61]. Five patients with COVID-19 were identified (confirmed *N* =  4, probable *N* =  1), and thus the prevalence of COVID-19 in these subjects was calculated to be 0.9% (95% CI, 0.1–1.8%). This was ninefold higher than that reported in healthy persons (0.1%) but lower than 10% as reported in hospitalized persons with other hematological cancers or in healthcare providers (7%, CI 4–12%). Subjects from that cohort exhibited an increased risk of developing COVID-19 if diagnosed with advanced phases of CML (*p*  =  0.004), even if they had achieved a complete cytogenetic response or major molecular response at the time of exposure to COVID-19. Covariates such as age and TKI therapy duration were not significantly associated with an increased risk of developing COVID-19.

The largest global cohort study so far characterizing COVID-19 in adult patients with CML was recently reported in an abstract form by the international CML foundation (iCMLf), which is a charitable foundation established to improve the outcomes of patients with CML globally [62]. One hundred ten cases of COVID-19 (median age: 54 years; range: 18–89) were collected from 20 countries. Among these, 91/110 cases were reported by physicians out of a total of 12,236 CML patients that they were treating (prevalence: 0.7%). COVID-19 was diagnosed by PCR or serology in 93 patients (85%) and clinically suspected in 17 patients (15%) while patients were under treatment (median duration: 7 years; range: 0–25) for CML with TKIs in most cases (hydroxyurea 1%, TKI 70%, 16% untreated at the time of COVID-19 diagnosis, 13% lacking information). During COVID-19 infection, 33 patients (30%) interrupted their TKI treatment, and 8/110 (7%) cases with COVID-19 were asymptomatic. In the 102 symptomatic patients (93%), the course of COVID-19 was considered to be mild (no hospitalization) in 49 cases (45%), moderate (hospitalization) in 19 cases (17%), severe (intensive care) in 19 cases (17%), and of unknown severity in 15 cases (14%). As of 1 July 2020, COVID-19 was still active in 14 patients (13%), and the outcome was unknown in 9 patients (8%), favorable in 75 patients (68%), and fatal in 12 patients (14%). The authors could show that factors associated with a higher mortality rate were older age and imatinib therapy. However, imatinib may represent a confounder effect, as a strong link between imatinib treatment and advanced age was identified.

An observational cohort study was conducted in the Netherlands during the COVID-19 pandemic [63] that assessed differences in susceptibility for COVID-19 and the severity of the disease’s course in adult CML patients (*N* = 148, median age: 57.5 years; range: 26–82), with their adult housemates (*N* = 123, median age: 60 years; range: 24–88) serving as the controls. In a preliminary report, no significantly increased prevalence of COVID-19 in adult CML patients was observed, and only one patient (0.7%) tested positive and required inpatient care.

In comparison with adults, the proportion of children without underlying diseases who are affected by COVID-19 is smaller. Reports so far have demonstrated that the severity of the infection’s course is mild, presenting as self-limiting symptoms of the upper respiratory tract in most children [64]. Only a very small number of children developed a rare multisystem inflammatory syndrome (MIS) or died from COVID-19. Data specifically in relation to outcomes in the pediatric oncology population are limited. An analysis of 33 studies from single centers and from national reports from different countries comprising a total of 226 children were comprehensively reviewed in a recent publication. The incidence of COVID-19 was found to be higher in children with neoplastic diseases than in the general pediatric population [65]. More children with hematological malignancies (*N* = 120) were affected than those with solid tumors (*N* = 76). As there was no analysis conducted with subgroups of hematological malignancies, it is unclear how many children with CML were included. In the entire cohort, the male gender and children in intensive treatment were affected more significantly, with fever as the leading symptom. The course of COVID-19 was asymptomatic or mild in 48% and severe in 9.6% of the children. Thirty-two percent needed oxygen support, 10% were admitted to the intensive care unit, and 4.9% died from COVID-19. In general, the severity, morbidity, and mortality of the infection in children with malignancies were more or less comparable to the general pediatric population. However, it must be kept in mind that the data sets are still small and heterogenous, and the findings in these studies vary as they are from different countries with diverse health infrastructures and policies. Thus far, there is no report on the course of COVID-19 in a child treated for CML.

#### 2.3.3. Vaccines against COVID-19

In the COVID-19 pandemic, various vaccines have been developed rapidly in different countries of the world (Table 1). Among these new types are mRNA-based vaccines (Comirnaty^®^, BioNTech/Pfizer, Mainz, Germany; Moderna COVID-19 Vaccine^®^, Moderna, Norwood, MA, USA) which had not been used in humans prior to the pandemic. Another new approach to vaccination in humans is represented by vector-based vaccines (COVID-19 Vaccine Janssen^®^, Janssen-Cilag, Beerse, Belgium; COVID-19 SARS-CoV2 Vaccine, Johnson & Johnson, New Brunswick, NJ, USA; Gam-COVID-Vac^®^/Sputnik V, Biocad, Moscow, Russia; Vaxzevria^®^, AstraZeneca, Nijmegen, The Netherlands). Both m-RNA and vector-based vaccines have been classified as inactivated vaccines.

The classical way of challenging the immune system with inactivated vaccines is represented by using either protein components (in this case, the spike (S)-protein of the COVID-19 virus) in conjunction with an adjuvant (NVX-CoV2372, Novavax, Gaithersburg, MD, USA) or by using inactivated strains (CN02, HB02) of SARS-CoV-2 in conjunction with Al(OH)_3_ as an adjuvant as produced and applied first as vaccines (CoronaVac, Sinovac Biotech, Being, China; BBIBP-CorV, Sinopharm ½) in China and also in India (Covaxin, Bharat Biotech, Genome Valley, Hyderabad, India).

The vaccines developed by different companies thus far are listed below (see Table 1). For a more detailed overview, we kindly refer the reader to an article by Creech et al. [66].

As of April 2021, trials for COVID-19 vaccines are also underway for children. For children with CML, the rarity of this type of leukemia represents an additional hurdle when trying to assess the benefit of a given vaccine. In the USA, the FDA approved the Comirnaty^®^ vaccine (Pfizer) for ages 12–15 on 20 May 2021. In ongoing trials, both Pfizer and Moderna started testing their COVID-19 vaccines in children aged 6 months to 11 years back in March 2021. This will further help to protect the community from passing the infection inadvertently at school and at home. In adults with CML, data are emerging that robust memory T-cell responses develop in patients with CML following infection with severe acute respiratory syndrome coronavirus-2 [67]. In a recent publication it was also shown in adults that COVID-19 vaccination induces immunity [68]. Sixteen patients with CML (median age: 48 years; range: 21–75 years) all developed neutralizing antibody responses, and 14/16 patients (87%) developed T-cell responses against SARS-CoV-2 infection 21 days following a single first dose of the Pfizer-BioNTech BNT162b2 vaccine. The vaccine was safe in this cohort, and tolerable side effects consisted of localized inflammation in 9 patients (56%) and a transient flu-like illness in 4 patients (25%). Four patients each were on treatment with imatinib, nilotinib, and ponatinib, with 2 patients each undergoing treatment with bosutinib and dasatinib. However, these encouraging preliminary results must be confirmed in further prospective trials.

## 3. Conclusions

TKI treatment for CML causes humoral and cellular immune dysfunction which is mild in most patients, and thus infectious complications are rare. Routine immunizations are important for the health maintenance of children, but vaccinations for children with CML on TKI therapy should be carefully considered. In general, inactivated vaccines are safe. There was a concern for the safety of live attenuated vaccines, but preliminary experience from a few recent case reports have shown that MMR vaccines could be administered safely. Indications of COVID-19 vaccination for children with CML do not differ from those for the general pediatric population.

## Figures and Tables

**Table 1 jcm-10-04056-t001:** SARS-CoV-2 vaccines developed and currently under emergency use application, modified from and, for more details, see [66].

Vaccine	Company	Vaccine Type	Antigen	Dose	Licensed or Emergency Use Application (EUA, as of 12 March 2021) in
Comirnaty^®^ Tozinameran BNT162b2	BioNtech (Germany). Pfizer (USA)	mRNA	Full-length spike (S) protein with proline substitutions of the SARS-CoV-2 virus	2 doses, each 30 µg, 21 days apart	Canada, EU, Japan, UK, USA *
Covid-19 Vaccine Moderna, (mRNA-1273)	Moderna (USA)	mRNA	Full-length spike (S) protein with proline substitutions of the SARS-CoV-2 virus	2 doses, each 100 µg, 28 days apart	Canada, EU, UK, USA
CVnCoV	Curevac (Germany), Bayer (Germany), Glaxo-Smith-Kline (UK)	mRNA	Prefusion stabilized full-length spike (S) protein with proline substitutions of the SARS-CoV-2 virus	2 doses, each 12 µg, 28 days apart	(under a rolling review in the EU)
Covid-19 Vaccine Astra-Zeneca, ChAdOx1, AZD1222	Astra-Zeneca (UK, Sweden)	Viral vector	Replication-deficient chimpanzee adenoviral vector with the SARS-CoV-2S protein	2 doses, each containing 5 × 10^10^ virus particles 28 days apart	Canada, EU, India, Mexico, UK
Covid-19 Vaccine Janssen, Ad26.CoV2.S	Janssen (Belgium), Johnson & Johnson (USA)	Viral vector	Recombinant replication incompetent human adenovirus-vector serotype 26, encoding a full-length, stabilized SARS-CoV-2 spike (S)-protein with proline-substitutions	1 dose containing 5 × 10^10^ virus particles	Canada, EU, USA
Gam-Covid-Vac, Sputnik V	Gamaleya National Research Center for Epidemiology and Microbiology (Russia)	Viral vector	Full-length recombinant spike (S) protein with proline substitutions of the SARS-CoV-2 virus, carried by each replication-incompetent human adenovirus vector serotype 5 or 26	2 doses (first with rAd26, second with rAd5), each containing 10^11^ virus particles 21 days apart	Algeria, Argentina, Belarus, Egypt, Palestina, Russia, Serbia (under a rolling review in the EU)
NVX-CoV2373	Novavax (USA)	Protein subunit	Recombinant full-length prefusion stabilized spike (S) protein	2 doses each, both containing 5 µg protein plus 50 µg matrix protein as adjuvants	(under a rolling review in the EU)
CoronaVac	Sinovac Biotech (China)	Inactivated virus	Inactivated strain CN02 of SARS-CoV-2, produced in Vero cells	2 doses, each containing 3 µg plus Al(OH)_3_ as adjuvants 14 days apart	Azerbaijan, Bolivia, Brazil, China, Chile, Columbia, Indonesia, Uruguay, Turkey
BBIBP-CorV	Sinopharm 1/2 (China)	Inactivated virus	Inactivated strain HB02 of SARS-CoV-2, produced in Vero cells	2 doses, each containing 4 µg plus Al(OH)_3_ as adjuvants 21 days apart	Bahrain, China, Peru, Serbia, Zimbabwe
Covaxin	Bharat Biotech (India)	Inactivated virus	Inactivated strain NIV-2020-770 of SARS-CoV-2, produced in Vero cells	2 doses, each containing 6 µg plus (Al(OH)_3_ as adjuvants at least 28 days apart	Guyana, India, Iran, Mauritius, Mexico, Nepal, Paraguay, the Philippines, Zimbabwe

* Note added during the reviewing process of the manuscript: Corminaty^TM^ received full approval by the FDA for individuals over 16 years old as of 23 August 2021.

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
