# Peer review of "Chronic Myeloid Leukemia in Children: Immune Function and Vaccinations"

_jcm, 2021, doi:10.3390/jcm10184056_

Round 1

Reviewer 1 Report

In this manuscript, Authors describe the effects of TKI on immune function and their implications for vaccination in children with chronic myeloid leukemia. Additionally, they touch upon a very important aspect of COVID-19 in these patients and analyze different vaccination options. 

My suggestions:

  1. abstract is too short, and there is a mistake in the following sentence: “TKI treatment for CML causes humoral and cellular immune dysfunction which is mild in most patients and thus infectious complications are.” The word “rare” should be at the end of this sentence.
  2. Section: 2.3.3 Vaccines against COVID-19, line 457: “ (Table)”, should be (Table 1).

Author Response

Response to Reviewer 1:

1) We thank Reviewer 1 for this suggestion. The length of the abstract text has been increased by 3-fold (from 66 to 201 words) in the revised version of the manuscript.

2) We thank Reviewer 1 for this comment, and we apologize for the mistake. The missing word “rare” was added to the sentence in the revised version of the manuscript

3) We apologize for this typing error. The missing figure “1” was added into the text of the revised version of the manuscript.

Reviewer 2 Report

Although CML in pediatric patient population is quite rare, the present review by Suttorp et al. focuses on defining the long-term effect of TKI and the need for vaccinations in children with CML. The review is comprehensively written and very informative, which should attract much attention of the field, especially in the current scenario during the COVID-19 epidemic. I have few comments which will help strengthen the review.

Major Comments:

  1. In Table 1, the authors must update that the Comirnaty (BNT162b2) vaccine was recently fully approved by the US FDA for COVID-19 in adult population.
  2. In Table 1, the authors must also include COVID-19 vaccine, COVAXIN developed by the company Bharat Biotech (India).
  3. Because the review is primarily focused on vaccinations among children with CML, another Table describing all the clinical trials currently being tested for children <12 years of age would further strengthen the review.

Author Response

Response to Reviewer 2

4) We thank Reviewer 2 for this comment. As a footnote we added the following text to Table 1 in the revised version of the manuscript:

“*) Note added during the reviewing process of the manuscript:

CorminatyTM received full approval by the FDA for individuals over 16 years old as of 23 Aug 2021.”

5) We apologize for having not mentioned the vaccine Covaxin produced by Bharat Biotech in India. We have added another line to Table 1 and filled in the necessary information.

6) We agree with Reviewer 2 that a table listing clinical trials currently being conducted in children <12 years of age testing vaccines against COVI-19 in children would further strengthen the review. However, to the best of our knowledge so far only Pfizer and Moderna have opened trials for kids below the age of 12 years old. This does not justify another table and therefore we have added the sentence “In ongoing trials, both Pfizer and Moderna started testing their COVID-19 vaccines in children aged six months to 11 years back in March 2021.” in line 497 of the revised version of the manuscript.